# Electroanatomical Mapping System-Guided vs. Intracardiac Echocardiography-Guided Slow Pathway Ablation: A Randomized, Single-Center Trial

**DOI:** 10.3390/jcm12175577

**Published:** 2023-08-26

**Authors:** Botond Bocz, Dorottya Debreceni, Kristof-Ferenc Janosi, Marton Turcsan, Tamas Simor, Peter Kupo

**Affiliations:** Heart Institute, Medical School, University of Pecs, Ifjusag Utja 13, H-7624 Pécs, Hungary; boczbotond@gmail.com (B.B.); debreceni.d@gmail.com (D.D.); janosikristof32@gmail.com (K.-F.J.); marcittm@gmail.com (M.T.); tsimor@hotmail.com (T.S.)

**Keywords:** electroanatomical mapping systems, intracardiac echocardiography, slow pathway ablation, AVNRT

## Abstract

Radiofrequency (RF) catheter ablation is an effective treatment option for targeting the slow pathway (SP) in atrioventricular nodal reentry tachycardia (AVNRT). Previous data suggested that using intracardiac echocardiography (ICE) guidance could improve procedural outcomes when compared to using fluoroscopy alone. In this prospective study, we aimed to compare the effectiveness of an electroanatomical mapping system (EAMS)-guided approach with an ICE-guided approach for SP ablation. Eighty patients undergoing SP ablation for AVNRT were randomly assigned to either the ICE-guided or EAMS-guided group. If the procedural endpoint was not achieved after 8 RF applications; patients were allowed to crossover to the ICE-guided group. The ICE-guided approach reduced the total procedure time (61.0 (56.0; 66.8) min vs. 71.5 (61.0; 80.8) min, *p* < 0.01). However, the total fluoroscopy time was shorter (0 (0–0) s vs. 83.5 (58.5–133.25) s, *p* < 0.001) and the radiation dose was lower (0 (0–0) mGy vs. 3.3 (2.0–4.7) mGy, *p* < 0.001) with EAMS-guidance. The ICE-guided group had a lower number of RF applications (4 (3–5) vs. 5 (3.0–7.8), *p* = 0.03) and total ablation time (98.5 (66.8–186) s vs. 136.5 (100.5–215.8) s, *p* = 0.02). Nine out of 40 patients (22.5%) in the EAMS-guided group crossed over to the ICE-guided group, and they were successfully treated with similar RF applications in terms of number, time, and energy compared to the ICE-guided group. There were no recurrences during the follow-up period. In conclusion, the utilization of ICE guidance during SP ablation has demonstrated notable reductions in procedural time and RF delivery when compared to procedures guided by EAMS. In challenging cases, an early switch to ICE-guided ablation may be the optimal choice for achieving successful treatment.

## 1. Introduction

Atrioventricular nodal reentrant tachycardia (AVNRT) is the predominant type of paroxysmal supraventricular tachycardia (SVT), which is characterized by reentry circuitry in the atrioventricular (AV) node region [1]. Although the precise circuitry underlying AVNRT remains unclear, existing evidence indicates that the inferior nodal extensions likely act as the substrate for conduction through the slow pathway (SP) [1].

The catheter ablation procedure targeting the slow pathway (SP) is widely recognized as a primary therapeutic approach for the management of AVNRT. It has consistently exhibited favorable outcomes with high success rates in both the short-term and long-term follow-up periods. However, in certain cases, the ablation of the SP can present challenges due to anatomical variations [2,3].

In SP ablation procedures, the placement of the ablation catheter is guided by anatomical considerations, and intracardiac electrograms are used to ensure accuracy [4,5]. Traditionally, fluoroscopy-guided catheter positioning was employed during the ablations. However, the advent of electroanatomical mapping systems (EAMS) over the last decades has demonstrated their superior efficacy compared to fluoroscopy-guided techniques, particularly in reducing fluoroscopy exposure during SP ablations [6,7,8].

Intracardiac echocardiography (ICE) is an exceptional imaging technique that provides real-time visualization of intracardiac structures, as well as the position and stability of the ablation catheter [9,10].

Previous research has shown that utilization of ICE guidance during SP ablation procedures results in a noteworthy reduction in mapping and ablation time, radiation exposure, and radiofrequency (RF) energy delivery. These findings highlight the advantages of using ICE guidance over fluoroscopy-only procedures [11].

Despite the utility of ICE and EAMS in SP ablation procedures for AVNRT, no scientific data currently exist comparing the efficacy of these two techniques. Consequently, our aim was to perform a study comparing the procedural outcomes of ICE-guided and EAMS-guided SP ablation in patients undergoing this procedure (Figure 1).

## 2. Methods

### 2.1. Study Population

In our single-center trial (registration number: NCT05907863), we recruited 80 patients consecutively who were scheduled to undergo an electrophysiology (EP) study and SP RF ablation for AVNRT. These patients were randomly assigned to two groups: ICE-guided or EAMS-guided ablation groups. The procedures were performed by an experienced electrophysiologist who possessed expertise and proficiency in utilizing both ICE and EAMS for SP ablations. Patients who were referred for a second procedure had other arrhythmias in addition to AVNRT or were under 18 years old were excluded from the study. The study was conducted in accordance with the Declaration of Helsinki and was approved by the regional ethics committee. All patients provided written informed consent to participate in the study.

### 2.2. Study Protocol

To prepare for the EP study, antiarrhythmic drugs were discontinued at least five half-lives before the procedure, and were under conscious sedation using midazolam ± fentanyl while fasting.

### 2.3. ICE-Guided Ablation Group

In patients randomized to the ICE-guided ablation group, catheter placement was initially performed using fluoroscopy guidance, after local anesthesia. A decapolar steerable catheter (ViaCath 10, Biotronik, Berlin, Germany) was placed in the coronary sinus (CS), a quadripolar electrode (Triguy, APTMedical, Shenzen, China) catheter was positioned in the right ventricular apex and an ablation catheter (Biotronik AlCath LT G FullCircle, Biotronik, Germany) was inserted to record the His bundle electrogram. During the study, twelve-lead electrocardiogram (ECG) and intracardiac electrograms were acquired and stored using a digital recording system. The recordings were filtered using a band pass filter with a frequency range of 30 to 500 Hz. In order to test AV nodal conduction and induce AVNRT, electrical stimulation techniques were employed.

A stepwise approach was followed during the study where the S2 coupling interval, which refers to the time interval between two consecutive electrical stimulations, was systematically decreased after each drive-train stimulation. The goal was to provoke specific outcomes, such as inducing tachycardia, observing AV conduction block, or reaching the refractory period of the atria. The process involved a gradual reduction in the S2 coupling interval until one of these predefined endpoints was achieved. In cases where tachycardia was not successfully induced during the initial stimulation protocol, an isoprenaline infusion was administered. The infusion aimed to raise the heart rate by a minimum of 20%. Following the infusion, the same stimulation protocol was repeated, encompassing both the infusion and subsequent washout phases.

The diagnosis of AVNRT was made using established electrophysiologic criteria and pacing maneuvers. The methods involved assessing the A-(H)-V response after ventricular overdrive pacing and measuring the SA-VA interval. A SA-VA interval greater than 85 ms was considered indicative of AVNRT. Additionally, the corrected postpacing interval minus the tachycardia cycle length was calculated, and a value greater than 110 ms supported the diagnosis of AVNRT [12,13].

After confirmation of the diagnosis of AVNRT through the diagnostic EP study, the quadripolar electrode catheter was removed and exchanged with an 8F ICE catheter (AcuNaV™ 90 cm, Siemens Medical Solutions, Mountain View, CA, USA) to facilitate both mapping and SP ablation. To enable clear visualization of the anatomical landmarks, the ICE was positioned within the low right atrium, specifically at the 6 o’clock position (Appendix A). Subsequently, the catheter was gently rotated in a clockwise direction towards the septum of the heart. This rotational movement ensured optimal imaging and identification of the relevant anatomical structures during the procedure. The distance between the ablation catheter and the compact AV node was assessed by measuring the distance from the aortic valve. The aortic valve serves as a reference point that indicates the recording site of a proximal His potential. This measurement technique helped determine the proximity of the ablation catheter to the compact AV node, ensuring accurate placement during the procedure. In cases of ineffective ablation, the catheter was moved closer to the aortic valve, but a distance of at least 0.5 cm was always maintained, and RF application was attempted again. RF energy was administered beginning just below the CS. The power output for RF energy delivery was set at 30 Watts, and a preset temperature of 55 °C was used as a target during the ablation process. Effective applications were continued for 30 to 60 s and considered successful when junctional rhythm appeared. In the event of catheter displacement, sudden impedance rise, prolongation of PR interval, anterograde AV block, or retrograde ventriculoatrial (VA) block, the delivery of RF energy was promptly ceased. These parameters were monitored closely during the procedure, and any of these signs or events indicated the need to halt the RF application to ensure patient safety and minimize potential complications.

### 2.4. EAMS-Guided Ablation Group

In the EAMS group, the operator’s intention was to carry out fluoroscopy-free procedures. An ablation catheter (NAVISTAR, Biosense Webster, CA, USA) was inserted into the heart to create an anatomical map by CARTO3 EAMS of the right atrium after local anesthesia, and the location of the His bundle was tagged. Decapolar and quadripolar diagnostic catheters were subsequently positioned in their appropriate locations as previously described. Notably, this was achieved without the use of fluoroscopy. Once the diagnosis of AVNRT was established, the mapping of the SP started using a NAVISTAR catheter guided by EAMS. The mapping was performed by assessing the atrial-to-ventricular electrogram amplitude ratio, with a desired range of 1:3 to 1:5. If the ablation endpoint was not reached after 8 radiofrequency (RF) applications, patients in the EMAS-guided ablation group were allowed to crossover to an ICE-guided procedure.

The ablation procedure was deemed successful if, following a 20-min waiting period, the arrhythmia failed to be induced and there were no instances of more than one echo beat observed, both in the presence and absence of isoprenaline.

The duration of the procedure was calculated by measuring the time elapsed from the initial femoral puncture, which marked the beginning of the intervention, until the withdrawal of the catheters, which indicated the conclusion of the procedure. The mapping plus ablation time was determined by measuring the duration starting from the initiation of the slow pathway (SP) mapping until the completion of the final attempted ablation. The fluoroscopy system automatically recorded the duration of fluoroscopy time, radiation dose, and dose-area product (DAP). The ablation data, including the total number of RF applications, cumulative RF energy delivered in watt-seconds (Ws), and the overall ablation time in seconds, were calculated and stored by the electrophysiology (EP) recording system (CardioLab, GE Healthcare, Chicago, IL, USA). The follow-up time was defined as the duration from the procedure to the last telemedicine ambulatory visit. All patients included in the study completed the full 1-year follow-up period. No instances of patient loss occurred during the study.

### 2.5. Statistical Analysis

The distribution pattern of the data was evaluated using Kolmogorov–Smirnov tests, which assessed the conformity of the data to a specific probability distribution. All statistical tests conducted were two-tailed, and a significance level of *p* < 0.05 was considered statistically significant. Continuous data were presented as mean ± standard deviation (SD) or median with interquartile range (IQR), depending on the appropriateness of the data distribution. Categorical variables were expressed as absolute numbers and percentages. For comparisons between groups, the chi-square test was used for categorical variables, while the T-test and Mann–Whitney U test were employed for continuous variables, depending on the nature of the data. All statistical analyses were performed using SPSS 24 software, developed by SPSS Inc. located in Chicago, IL, USA.

## 3. Results

We enrolled 80 patients in our study, with 40 patients assigned to the EAMS-guided group and 40 patients to the ICE-guided SP ablation group. There were no significant differences in baseline characteristics, including sex (female: 70.0% vs. 67.5%, *p* = 0.81) and age (49.6 ± 14.9 vs. 53.0 ± 13.4 years, *p* = 0.15), between the groups (Table 1). All 80 cases achieved the procedural endpoint, resulting in a 100% acute success rate. However, the ICE-guided group had significantly shorter procedure time (61.0 (56.0; 66.8) min vs. 71.5 (61.0; 80.8) min, *p* < 0.01), puncture to mapping time (32.5 ± 8.5 min vs. 40.3 ± 10.2 min, *p* < 0.01), and mapping plus ablation time (3 (2; 10.25) min vs. 6.5 (3; 20.5) min, *p* = 0.04).

In contrast, the EAMS-guided group had significantly lower total fluoroscopy time (0 (0–0) s vs. 83.5 (58.5–133.25) s, *p* < 0.001) and total fluoroscopy exposure (0 (0–0) mGy vs. 3.3 (2.0–4.7) mGy, *p* < 0.001) due to the absence of fluoroscopy use during the procedures (no fluoroscopy use was required). Furthermore, after inserting the ICE catheter in the ICE-guided group, fluoroscopy was not necessary. The total ablation time (98.5 (66.8–186) s vs. 136.5 (100.5–215.8) s, *p* = 0.02) and the number of RF applications (4 (3–5) vs. 5 (3.0–7.8), *p* = 0.03) were also lower in the ICE-guided group, although no significant difference was found in the sum of delivered RF energy (3052 (2027–5070) Ws vs. 3572 (2591–6210) Ws, *p* = 0.16) between the two groups.

In the EAMS-guided group, a total of 9 out of 40 patients (22.5%) were crossed over to the ICE-guided group based on the operator’s discretion, as specified in the methods section. This decision was made due to the failure to achieve the ablation endpoint after 8 RF applications: despite a favorable response to the ablation (junctional acceleration) in 5 cases, the original arrhythmia remained inducible. In 3 cases, junctional acceleration could not be achieved. Following the crossover, in 4 cases, a steerable sheath was introduced to enhance the stability of the ablation catheter. After the crossover, all patients were treated successfully with similar RF applications in terms of number, time, and cumulative energy compared to the ICE-guided group, as shown in Table 2. No complications occurred during the study, and there were no instances of recurrence during the 12.8 ± 3.2-month follow-up period.

## 4. Discussion

In this randomized comparative study evaluating the outcomes of EAMS-guided versus ICE-guided ablation of the SP for AVNRT, our findings indicate that the use of ICE is advantageous in reducing the duration of mapping and ablation procedures, minimizing unnecessary RF energy delivery, and enabling successful treatment of complex cases.

The conventional method for ablation of the SP using fluoroscopy guidance is highly effective but poses significant risks related to radiation exposure for both the operating personnel and the patients. Continuous exposure to radiation can result in increased risks of cataracts, dermatitis, and cancer due to both stochastic and deterministic effects [14,15,16]. To reduce the above-mentioned risk of fluoroscopy, in recent years different zero/minimal-fluoroscopic (Z/MF) techniques have been developed.

EAMS utilizes specialized software to track the position of catheters within the heart, providing a 3D map of the atrial cavity that can be manipulated for optimal visualization. This technology can provide information regarding the location and depth of the applied lesions, as well as details about components of the conduction system (e.g., His bundle). However, it should be noted that the ability of EAMS to accurately represent anatomic variations is limited [7,17,18].

The implementation of EAMSs in ablation procedures enables the use of a Z/MF strategy, thereby eliminating radiation hazards for both patients and personnel. Multiple studies have previously conducted comparisons between the Z/MF fluoroscopy strategy and the conventional approach in the treatment of AVNRTs [7,19,20]. Studies have shown, that catheter ablation for AVNRT without fluoroscopic guidance is feasible and safe, and does not prolong procedure time. A recent meta-analysis that included 24 studies and 9074 patients compared the efficacy of Z/MF and conventional approaches in treating SVT. The analysis found that the use of EAMS significantly reduced radiation dose, fluoroscopy time, and ablation time. However, EAMS guidance had no significant impact on total procedural time, acute and long-term success rates, or complication rates [21].

In certain situations, the utilization of steerable sheaths may be necessary during SVT ablations. However, this has previously posed a challenge to the application of the Z/MF strategy. The recent development of visualizable steerable sheaths which can be tracked by EAMSs, has improved procedural outcomes by allowing for the implementation of the Z/MF strategy even in cases where steerable sheaths are required [22].

Although EAMSs can effectively decrease both fluoroscopy time and dose, they may not be as useful in identifying anatomical variations since they lack direct visualization of anatomical structures. ICE is an advanced imaging modality that enables the dynamic visualization of intracardiac structures in real time. The ICE-guided method provides a clear and immediate visual representation of the heart, allowing for the detection of potential anatomic variations within the Koch-triangles area. This area is known to vary significantly in terms of both anatomy and electrophysiology between individuals [23,24].

According to a previously published randomized trial, ICE-guided SP ablation has been found to have advantages over the conventional fluoroscopic method. Specifically, the use of ICE guidance resulted in reduced total fluoroscopy time and radiation exposure, as well as a reduction in total ablation time, energy requirements, and the number of necessary ablation applications. Notably, due to the study protocol, a quarter of the patients in the fluoroscopy group switched to the ICE-guided group. Nevertheless, all patients were effectively treated exhibiting comparable numbers of RF applications, durations of the procedure, and cumulative RF energy delivery in comparison to the ICE group [11].

Consistent with these findings, our study has provided evidence that integrating ICE guidance during RF ablation of the SP not only reduces the need for radiation exposure during the procedure but also reduces unnecessary RF energy delivery. This is achieved by offering a clear visualization of both the ablation catheter and the targeted SP region, enabling more precise and targeted ablation without the reliance on fluoroscopy.

The observed decrease in RF energy delivery may potentially influence the occurrence of late conduction disturbances following the procedure [25]. Remarkably, even in challenging procedures initially guided by EAMS but subsequently crossed over ICE guidance, comparable quantities, durations, and cumulative energy of RF applications were necessary when compared to the group guided solely by ICE. The crossover rate to ICE in the EAM-guided cases was observed to be 22.5%. This crossover rate is comparable to the rate found in a previous study that compared fluoroscopy-guided and ICE-guided SP ablations [11]. This implies that using ICE to directly visualize the target area and catheter during SP ablation is just as effective, even in cases of unusual anatomy. Importantly, four individuals successfully completed the procedure using a steerable catheter. This observation suggests that the efficacy of the procedure might be more closely linked to the stability of the catheter rather than the specific guidance method employed, whether it be EAM or ICE.

In our opinion, the decision to employ a steerable sheath can be expedited through the use of ICE, as the visualization offered by ICE distinctly reveals instances of unstable catheter–tissue contact. Such scenarios, if uncorrectable through conventional methods, may necessitate the use of a steerable sheath.

It is noteworthy to emphasize that the utilization of ICE in the context of AVNRT ablation bestows a unique advantage through its ability to enhance the visualization of intracardiac structures, thereby augmenting procedural precision. However, it is imperative to duly recognize that intracardiac signals hold a pivotal role in guiding the accurate positioning of the catheter during the procedure.

The sole potential drawback to using an ICE-guided approach for SP ablation is the increased cost associated with the utilization of the catheter. However, this expense is comparable to or even lower than the cost of using an EAMS, especially when using reprocessed ICE catheters [26,27]. In comparison to an EAMS, ICE may offer the additional benefit of more precise ablation targeting with reduced energy delivery, as well as facilitating treatment of more complex cases.

## 5. Limitations

The applicability of this study may be limited to the specific center where it was conducted, as it was a single-center study with a limited number of patients. Furthermore, due to the lack of blinding, the results may have been subject to bias. Noteworthy is that the study protocol did not aim to primarily demonstrate the difference in fluoroscopy use between the ICE and EAMS groups, as ICE was used exclusively for mapping and ablation purposes. All procedures were performed by a single operator who had experience and familiarity with the use of ICE for SP ablations. We acknowledge that our center has accumulated substantial expertise in ICE-guided procedures, which might have contributed to the observed outcomes. This expertise may have influenced the outcomes and procedural success rates observed in the study. Therefore, the results may not necessarily reflect the outcomes that would be obtained by operators with varying levels of experience or expertise in ICE-guided procedures. These limitations should be taken into consideration when interpreting the findings of our study.

## 6. Conclusions

Utilizing ICE guidance for anatomical SP ablation presents notable advantages over EAMS-guided procedures. These benefits include decreased mapping and ablation time, as well as reduced RF energy delivery. When EAMS-guided ablation is unsuccessful despite a reasonable attempt, transitioning to ICE-guided ablation can be a recommended alternative.

## Figures and Tables

**Figure 1 jcm-12-05577-f001:**
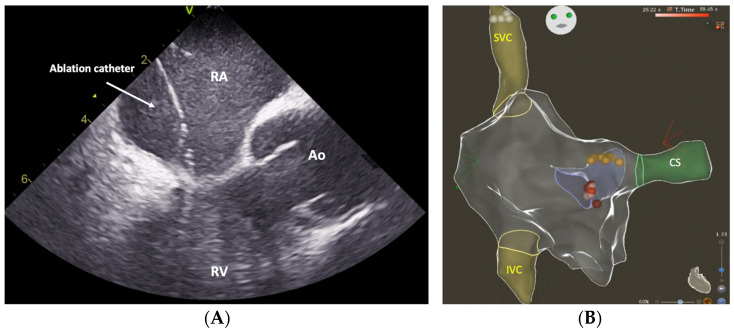
Panel (**A**): Intracardiac echocardiography (ICE) enables the direct observation of the ablation catheter within the region of the slow pathway. Panel (**B**): Three-dimensional electroanatomical map for slow pathway ablation. Abbreviations: Ao, aortic root; CS, coronary sinus; IVC, inferior vena cava; RA, right atrium; RV, right ventricle; and SVC, superior vena cava.

**Table 1 jcm-12-05577-t001:** Procedural parameters in the study population.

	ICE-Guided	EAMS-Guided	*p*-Value
Puncture to mapping time (min)	32.5 ± 8.5	40.3 ± 10.2	<0.001
Mapping plus ablation time (min)	3 (2; 10.25)	6.5 (3; 20.5)	0.04
Total procedure time (min)	61 (56.0; 66.8)	71.5 (61.0; 80.8)	0.004
Fluoroscopy time (s)	83.5 (58.5; 133.25)	0 (0; 0)	<0.001
Fluoroscopy dose (mGy)	3.3 (2.0; 4.7)	0 (0; 0)	<0.001
Radiation exposure (Gycm2)	0.41 (0.27; 0.72)	0 (0; 0)	<0.001
Number of RF applications	4 (3; 5)	5 (3; 7.8)	0.03
Total ablation time (s)	98.5 (66.8; 186.0)	136.5 (100.5; 215.8)	0.02
Sum of delivered energy (Ws)	3052 (2027; 5070)	3572 (2591; 6209.5)	0.16
Fluoroscopy time from diagnosis to the end of the procedure (s)	0 (0; 0)	0 (0; 0)	1
Fluoroscopy dose from diagnosis to the end of the procedure (mGy)	0 (0; 0)	0 (0; 0)	1
Radiation exposure from diagnosis to the end of the procedure (Gycm2)	0 (0; 0)	0 (0; 0)	1

RF, Radiofrequency; ICE, Intracardiac echocardiography; EAMS, Electroanatomical mapping system.

**Table 2 jcm-12-05577-t002:** Procedural parameters in the crossover group.

	Crossover Group
Fluoroscopy time from CO (min)	0 (0; 0)
Fluoroscopy dose from CO (mGy)	0 (0; 0)
Radiation exposure from CO (Gycm2)	0 (0; 0)
Ablation time from CO (s)	103 (90; 122)
Number of RF applications from CO	4 (3; 6)
Sum of delivered energy from CO (Ws)	2673 (2100; 3368)

CO, Crossover; RF, Radiofrequency.

## Data Availability

The data presented in this study are available on request from the corresponding author. The data are not publicly available due to Hungarian legal regulations.

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
