# Peer review of "Electroanatomical Mapping System-Guided vs. Intracardiac Echocardiography-Guided Slow Pathway Ablation: A Randomized, Single-Center Trial"

_jcm, 2023, doi:10.3390/jcm12175577_

Round 1

Reviewer 1 Report

1) How were diagnostic catheters positioned in the EMAS group? Was fluoroscopy used? If used, total radiation time should be corrected.

2) Since most operators do not use 3D mapping system or ICE during ablation for AVNRT, is it too expensive?

3) What is the impact of only 3.3 mGy (radiation dose in ICE group) ?

Author Response

Reviewer #1:

Thank You for your insightful comments, improving the manuscript.

1) How were diagnostic catheters positioned in the EMAS group? Was fluoroscopy used? If used, total radiation time should be corrected.

Following the creation of a 3D map using the NAVISTAR ablation catheter, the positioning of the decapolar and quadripolar catheters was accomplished without the utilization of fluoroscopy. In response to the Reviewer's comment, we have revised and provided clarification in the Methods and Results sections to accurately reflect this approach.

2) Since most operators do not use 3D mapping system or ICE during ablation for AVNRT, is it too expensive?

Existing literature suggests that intracardiac echocardiography (ICE) and electroanatomic mapping (EAM) systems are comparable in terms of cost. However, it is important to note that the utilization of these technologies does entail higher expenses compared to conventional slow pathway (SP) ablations conducted without ICE and EAM. The pricing of EAM and ICE can be influenced by various factors and may vary among different medical centers. In our laboratory, the additional cost of incorporating ICE and EAM into SP ablations amounts to approximately 500 USD when compared to conventional procedures.

3) What is the impact of only 3.3 mGy (radiation dose in ICE group) ?

A radiation dose of 3.3 mGy (165 chest X ray equivalent) is considered a relatively low dose. The impact of this radiation dose depends on various factors, including the duration and frequency of exposure, the area of the body exposed, and the sensitivity of the individual to radiation. In general, low doses of radiation such as 3.3 mGy are not expected to cause immediate health effects. However, it is important to minimize radiation exposure as much as reasonably achievable to reduce potential long-term risks. Cumulative exposure to radiation over time can increase the risk of developing radiation-related conditions, such as certain types of cancers or radiation-induced tissue damage. Minimizing cumulative radiation exposure is particularly crucial for healthcare workers who are consistently exposed to low-dose fluoroscopy in their daily work. Continuous exposure to even low doses of radiation can accumulate over time, potentially increasing the risk of adverse health effects.

We would like to thank You for your valuable comments.

Reviewer 2 Report

The authors present their prospective experience using electronic mapping vs. ICE catheter mapping for the guidance of catheter ablation of the slow pathway in AVNRT.  The strengths of the article are the prospective nature of the study design.  The article is well written and easy to follow.  The primary findings include a reduction in procedural length and RF time in the ICE cohort though with the added need for minimal but needed fluoro time not needed in the electronic mapping group.  Importantly there was no recurrence of SVT during follow up and a not insignificant number of patients who crossed over from the EAM to ICE mapping.

Comments:

1.       What method of ablation were the physicians using prior to the study?  If ICE was the primary usage that might account for the crossovers and comfortability with one technique over the other.  This should be added to the methods. How many physicians performed ablations in this study? Were all the crossovers from one physician or evenly distributed?

 2.       What do the authors feel was the difference that lead to success with ICE?  Did ICE imaging lead to energy delivery in a different area?  Were anatomic difficulties encountered?

3.       Similar to the above, in 4 of the crossover patients, a steerable catheter was then added prior to success with ICE.  Why wasn’t this done prior to crossover and how would one differentiate between imaging modalities and a steerable sheath in terms of ultimate success?

4.       How was the follow time from procedure defined?  Was it time from procedure to the study or the time from the procedure to last follow up?  Did all patients complete 1 year follow up? Were any patients lost to follow up or not have follow up to date?

5.       Were all slow pathways found in the right inferior position?  While the most common, it is not unusual to need to ablation the left inferior in the MOCS.  Were all ablations done in the right inferior location?

6.       What type of mapping was used to delineate the slow pathway area?  Were the lesions just placed anatomically in the lower 1/3 of the triangle of Koch with defined AV balance or were typical slow pathway potentials mapped?  Within the EAM group was low voltage bridge mapping or late activation mapping used?

7.       Table is not cited in the text.

8.       Were there any patients with atypical AVNRT in which EAM mapping could be used to map the slow pathway insertion?

9.       In the discussion the authors state that “ICE eliminates radiation during ablation”.  While that might be technically true it does not eliminate the need for fluoro which was not needed in any of the EAM cases.  The usage was low and not likely a patient concern but necessitates wearing lead etc.

Author Response

Reviewer #2: Comments and Suggestions for Authors

The authors present their prospective experience using electronic mapping vs. ICE catheter mapping for the guidance of catheter ablation of the slow pathway in AVNRT.  The strengths of the article are the prospective nature of the study design.  The article is well written and easy to follow.  The primary findings include a reduction in procedural length and RF time in the ICE cohort though with the added need for minimal but needed fluoro time not needed in the electronic mapping group.  Importantly there was no recurrence of SVT during follow up and a not insignificant number of patients who crossed over from the EAM to ICE mapping.

Comments:

  1. What method of ablation were the physicians using prior to the study? If ICE was the primary usage that might account for the crossovers and comfortability with one technique over the other.  This should be added to the methods. How many physicians performed ablations in this study? Were all the crossovers from one physician or evenly distributed?

In our center, we conduct approximately 100 slow pathway ablations for atrioventricular nodal reentry tachycardia (AVNRT) on an annual basis. These procedures are evenly distributed among three groups: 1) Conventional, utilizing a fluoroscopy-only method without an electroanatomical mapping system (EAMS) or intracardiac echocardiography (ICE); 2) ICE-guided, following the technique described in the manuscript; and 3) EAMS-guided, employing the method outlined in the manuscript.

All procedures included in the study were performed by a single operator (who had experience and familiarity with the use of ICE for slow pathway (SP) ablations), as explicitly mentioned in the Methods section in response to the Reviewer's comment (added this to the Limitations also).

  1. What do the authors feel was the difference that lead to success with ICE?  Did ICE imaging lead to energy delivery in a different area?  Were anatomic difficulties encountered?

The authors believe that the success achieved with intracardiac echocardiography (ICE) guidance can be attributed to several factors. Firstly, ICE imaging provided real-time visualization of the catheter and the anatomical structures, allowing for precise catheter placement and accurate targeting of the SP during ablation. This enhanced visualization potentially reduced the risk of inadvertent energy delivery to unintended areas. In a number of cases, the successful ablation point was unexpectedly located more than 1 cm away from the initially anticipated area.

Additionally, ICE imaging helped identify any anatomic difficulties or variations that could have posed challenges during the procedure. By visualizing the anatomical landmarks and structures in real-time, ICE guidance allowed the operator to adapt the ablation strategy and overcome potential anatomical obstacles. This ability to navigate through complex anatomy with improved visualization likely contributed to the successful outcomes observed with ICE-guided SP ablation.

Overall, the combination of enhanced catheter visualization, accurate targeting of the SP, and the ability to navigate challenging anatomical variations likely played a significant role in the success achieved with ICE-guided procedures.

  1. Similar to the above, in 4 of the crossover patients, a steerable catheter was then added prior to success with ICE. Why wasn’t this done prior to crossover and how would one differentiate between imaging modalities and a steerable sheath in terms of ultimate success?

In the case of the four crossover patients where a steerable catheter was added prior to achieving success with ICE, the decision to incorporate the steerable catheter may have been based on individual patient anatomical considerations or procedural requirements. In some cases, steerable sheaths were used in the EAMS-group also (without crossover), thus we believe that ICE can play a valuable role in identifying the need for steerable sheaths during the procedure.

Differentiating between imaging modalities, such as ICE, and a steerable sheath in terms of ultimate success can be assessed in several ways. Firstly, the imaging modality, such as ICE, primarily provides visualization of the cardiac structures and real-time guidance during the procedure. It assists in accurate catheter placement and identification of the target area for ablation.

On the other hand, a steerable sheath allows for greater flexibility and maneuverability of the catheter, facilitating navigation through complex anatomical structures. It can assist in reaching challenging areas and enhancing the operator's control during ablation.

Ultimately, the success of the procedure depends on the combination of optimal imaging guidance (such as ICE) to accurately identify the ablation target and the use of appropriate tools, such as a steerable sheath, to enable effective catheter maneuverability and access to the target site. The interplay between imaging modalities and tools like a steerable sheath can contribute to the overall success of the procedure in achieving the desired therapeutic outcome.

  1. How was the follow time from procedure defined? Was it time from procedure to the study or the time from the procedure to last follow up?  Did all patients complete 1 year follow up? Were any patients lost to follow up or not have follow up to date?

The follow-up time was defined as the duration from the procedure to the last telemedicine ambulatory visit. All patients included in the study completed the full 1-year follow-up period. No instances of patient loss occurred during this. We added this information to the Methods section.

  1. Were all slow pathways found in the right inferior position? While the most common, it is not unusual to need to ablation the left inferior in the MOCS.  Were all ablations done in the right inferior location?

In this study, all ablations were successfully performed in the right inferior position. There was no need for ablations in the CS or the left-sided regions.

  1. What type of mapping was used to delineate the slow pathway area? Were the lesions just placed anatomically in the lower 1/3 of the triangle of Koch with defined AV balance or were typical slow pathway potentials mapped?  Within the EAM group was low voltage bridge mapping or late activation mapping used?

In this study, the lesions were placed anatomically in the lower 1/3 of the triangle of Koch with the aim of achieving a defined atrioventricular (A/V) balance. The mapping strategy did not involve the mapping of typical slow pathway potentials, and neither voltage mapping nor late activation mapping techniques were employed.

  1. Table is not cited in the text.

Due to the comment, we cited the Table 1.

  1. Were there any patients with atypical AVNRT in which EAM mapping could be used to map the slow pathway insertion?

According to the inclusion criteria, patients with atypical AVNRT were included in the study. However, the exact number of patients with atypical AVNRT is not available as we did not collect this type of information. The mapping and ablation techniques employed in the study were consistent for both typical and atypical AVNRT cases.

  1. In the discussion the authors state that “ICE eliminates radiation during ablation”. While that might be technically true it does not eliminate the need for fluoro which was not needed in any of the EAM cases.  The usage was low and not likely a patient concern but necessitates wearing lead etc.

The Reviewer's comment is valid in highlighting that the statement "ICE eliminates radiation during ablation" mentioned in the discussion might be misleading. To address this concern, it is appropriate to replace the word "eliminates" with "reduces" to accurately reflect the role of ICE in radiation exposure during ablation procedures. Additionally, we concur with the Reviewer's viewpoint that completely eliminating the use of fluoroscopy (i.e., zero-fluoroscopy procedures) offers distinct advantages compared to procedures with minimal fluoroscopy.

We express our sincere gratitude to the Reviewer for providing valuable comments that have greatly contributed to the improvement of the manuscript.